# Acute Effects of Focused Ultrasound-Induced Blood-Brain Barrier Opening on Anti-Pyroglu3 Abeta Antibody Delivery and Immune Responses

**DOI:** 10.3390/biom12070951

**Published:** 2022-07-06

**Authors:** Praveen Bathini, Tao Sun, Mathias Schenk, Stephan Schilling, Nathan J. McDannold, Cynthia A. Lemere

**Affiliations:** 1Ann Romney Center for Neurologic Diseases, Brigham and Women’s Hospital, Harvard Medical School, 60 Fenwood Road, Boston, MA 02115, USA; pbathini@bwh.harvard.edu; 2Focused Ultrasound Laboratory, Department of Radiology, Brigham and Women’s Hospital, Harvard Medical School, 75 Francis Street, Boston, MA 02115, USA; taosun@bwh.harvard.edu; 3Department of Molecular Drug Biochemistry and Therapy, Fraunhofer Institute for Cell Therapy and Immunology, Weinbergweg 22, 06120 Halle, Germany; mathias.schenk@izi.fraunhofer.de (M.S.); stephan.schilling@izi.fraunhofer.de (S.S.); 4Faculty of Applied Biosciences and Process Technology, Anhalt University of Applied Sciences, Bernburger Strasse 55, 06366 Kothen, Germany

**Keywords:** focused ultrasound, pyroglutamate-3 Aβ, microglia

## Abstract

Alzheimer’s Disease (AD) is a neurodegenerative disorder characterized by the accumulation of amyloid plaques and hyperphosphorylated tau in the brain. Currently, therapeutic agents targeting amyloid appear promising for AD, however, delivery to the CNS is limited due to the blood-brain-barrier (BBB). Focused ultrasound (FUS) is a method to induce a temporary opening of the BBB to enhance the delivery of therapeutic agents to the CNS. In this study, we evaluated the acute effects of FUS and whether the use of FUS-induced BBB opening enhances the delivery of 07/2a mAb, an anti-pyroglutamate-3 Aβ antibody, in aged 24 mo-old APP/PS1dE9 transgenic mice. FUS was performed either unilaterally or bilaterally with mAb infusion and the short-term effect was analyzed 4 h and 72 h post-treatment. Quantitative analysis by ELISA showed a 5–6-fold increase in 07/2a mAb levels in the brain at both time points and an increased brain-to-blood ratio of the antibody. Immunohistochemistry demonstrated an increase in IgG2a mAb detection particularly in the cortex, enhanced immunoreactivity of resident Iba1+ and phagocytic CD68+ microglial cells, and a transient increase in the infiltration of Ly6G+ immune cells. Cerebral microbleeds were not altered in the unilaterally or bilaterally sonicated hemispheres. Overall, this study shows the potential of FUS therapy for the enhanced delivery of CNS therapeutics.

## 1. Introduction

Alzheimer’s disease (AD), the most common form of dementia, is characterized by extracellular deposition of amyloid-β (Aβ) plaques, hyperphosphorylated tau in the neurofibrillary tangles, and neuroinflammation [1]. Aβ starts accumulating 15–20 years or more before the clinical symptoms appear and is considered a key factor in AD pathogenesis [2]. Various agents targeting Aβ in experimental mouse AD models and human AD subjects are under investigation and are a potential therapeutic strategy for preventing and treating the disease. Both active and passive immunizations are being explored and are at different stages of clinical trials [3]. The recent approval of aducanumab is under debate in the scientific community and the improvement in cognition was statistically significant in only a subset of patients who received the highest dose of aducanumab, a fully human IgG1 anti-amyloid monoclonal antibody (mAb) that binds plaque amyloid [4], while a subset of patients developed Amyloid related Imaging abnormalities (ARIA) with mostly transient and asymptomatic vasogenic edema and/or microhemorrhages [5,6,7]. Although controversial, the aducanumab clinical trials demonstrated that higher antibody exposure was associated with better biomarker changes and clinical outcomes [8].

Pyroglutamate-3 Aβ (pGlu-3 Aβ), an N-terminally truncated and modified form of Aβ, is highly neurotoxic with an enhanced propensity for aggregation and neurotoxicity compared to full-length Aβ peptides, has a potential role in seeding Aβ oligomerization and accumulation, making a target for immunotherapy studies [9,10]. Donanemab, another promising antibody, targets this specific form of amyloid in the brain, however, is also associated with ARIA, although to a lesser extent compared to Aducanumab [11]. However, most immunotherapies will likely require treatment of patients with high doses over many months to allow CNS penetration, thereby raising questions about the safety and side effects of the treatment [12]. Thus, the need for new approaches is of utmost importance in aiding drug delivery and plaque clearance with minimal side effects. While passive diffusion of antibodies across BBB is mediated by (a) adsorptive-mediated endocytosis (b) carrier-mediated transport and (c) receptor-mediated transcytosis [13], it has been estimated that only low doses (0.1%) of antibodies administered peripherally reach the brain, because of the difficulties associated with crossing the blood-brain barrier (BBB) [14,15]. Increased transfer of therapeutic agents across BBB can be achieved by non-invasive strategies through optimizing the biochemical properties of the compound [16], use of transport mechanisms [17], and employing techniques like ultrasound, hyperosmotic solutions to disrupt the tight junctions of the vasculature [18,19,20]. Therapeutic focused ultrasound (FUS) when coupled with the contrast agent microbubbles (MB) is known to cause transient BBB opening to facilitate the entry of therapeutic agents into the brain without signs of cellular damage and microhemorrhages [21]. Ultrasound induced MB oscillations generate shear stress and radiating forces at the endothelial surface, affecting the integrity of the tight junction endothelial proteins [22]. Safety studies showed that frequent exposure to sufficient levels of FUS along with MB infusion is safe and poses minimal risk to brain tissue [23]. Pilot studies in humans with mild AD demonstrated the safety of the treatment after repeated BBB disruption using an implantable ultrasound device. Although a non-significant lowering of amyloid accumulation was observed, the study demonstrates the potential role of ultrasound therapy in AD [24]. In a world-first trial, MRI-guided focused ultrasound was successfully used to investigate the feasibility and safety of opening BBB in patients with amyotrophic lateral sclerosis [25]. Experimental studies in rodents suggested that repeated opening of BBB through scanning ultrasound helped plaque removal with signs of microglial activation in AD mouse models [26]. Importantly, ultrasound treatment can improve drug targeting, reduce systemic overdose, and serve as a supplemental treatment along with amyloid immunotherapy.

In preclinical studies, ultrasound treatment in combination with therapeutics like murine chimeric IgG2a aducanumab analog [27] and a GSK-3 inhibitor [28] has been shown to increase brain levels of the therapeutic agents and plaque clearance. FUS enhanced the delivery of Intravenous immunoglobulin (IVIg), a therapeutic used to treat various neurologic diseases, thereby promoting hippocampal neurogenesis [29]. In a mouse model of AD, FUS was used to enhance the delivery of D3, a TrkA agonist to restore the neuroprotective function of neurons [30]. Interestingly, some studies showed that low-intensity ultrasound exposure without any opening of BBB can increase synaptic signaling, neurogenesis and restore cognitive function in aged mice [31]. Previously, we showed that the anti-pGlu-3 Aβ IgG2a mAb (07/2a) mAb lowered pGlu-3 Aβ and general Aβ and improved cognition in APP/PS1dE9 mice treated from 12 to 16 months of age [32]. In our recent study, we showed that three weekly doses of FUS-BBBD treatment in combination with murine 07/2a mAb in 16 mo-old APP PS1dE9 AD mice reduced pGlu-3 Aβ and Aβ42 plaque load and increased glial activity, resulting in significant improvement in cognition [33]. Controls included PBS alone, mAb alone and FUS alone. While the impact of repeated sonication’s was studied, it is imperative to understand the acute effects following sonication. In the present study, we sought to determine the acute temporal response to the FUS and anti-pGlu-3 Aβ combination treatment in the brain with emphasis on microglial activation and peripheral immune cell response following ultrasound and anti-pyroglutamate3Aβ treatment.

## 2. Materials and Methods

### 2.1. Animals

The experiments were carried out using a group of 24 months old (mo-old) APP/PS1 male and female mice in one study (unilateral FUS) and another group of 24 mo-old APP/PS1 male mice for a second study (bilateral FUS; described below). Mice were housed five per cage with constant room temperature (22 ± 3 °C) and 12 h light-dark cycle with free access to food and water. An equal number of mice were randomly assigned to treatment and control groups. Procedures involving animals and their care were approved and conducted in conformity with the institutional guidelines (IACUC, Institutional Animal Care and Use Committee) at Brigham and Women’s Hospital and Harvard Medical School as per the Guide for the Care and Use of Laboratory Animals. The assurance of compliance is A3431-01 for Harvard Medical School and A4752-01 for Brigham and Women’s Hospital, on file with the Office of Laboratory Animal Welfare (OLAW).

### 2.2. Treatment

FUS was performed as per the prescribed protocol [33] at the Department of Radiology, Brigham and Women’s Hospital. Briefly, immediately before the ultrasound exposure, mice were anesthetized with ketamine (80 mL/kg/h) and xylazine (10 mL/kg/h) intraperitoneal injection. Before FUS-BBBD, the mouse’s scalp was shaved, hair cleared with depilatory cream, a catheter was inserted into the tail vein, and mice were placed in a stereotactic frame. Mice were laid in the supine position and maintained under isoflurane anesthesia before FUS-BBBD or murine anti-pGlu-3 Aβ IgG2a mAb (07/2a) administration. A total of 29 mice including controls were assigned to two studies (Figure 1A). In the first study, “unilateral FUS”, male and female mice received 300 µg of 07/2a mAb by i.v. infusion followed immediately by contrast agent microbubble (MB) i.v. infusion and cortical FUS-BBBD (10-ms bursts/2 Hz/100 s) targeting the right hemisphere with the contralateral left hemisphere serving as control. These AD-like mice show significant plaque pathology in the cortex and hippocampus at 24 months of age and in this study, FUS-BBBD was focused on the cortical area. Mice were euthanized either at 4 h (*n* = 5/group) or 72 h (*n* = 4/group) after mAb infusion and received 2% trypan blue dye before euthanasia. In the second study, “bilateral FUS”, male mice were i.v. infused with a single dose of 300 μg 07/2a mAb followed by microbubble i.v. infusion with or without bilateral cortical FUS-BBBD (10-ms bursts/2 Hz/100 s) and euthanized 4 or 72 h later (*n* = 5/group × 4 groups). Control mice were treated with mAb alone. Upon euthanasia, blood samples were taken from the right cardiac ventricle, and mice were transcardially perfused. Brains were harvested and bisected into two hemispheres except for the unilateral FUS mice which were postfixed in 4% PFA for 24 h at 4 °C. For the bilateral FUS treated mice, left hemibrains were collected and stored at −80 °C. All fixed hemispheres were transferred into 1× PBS solution containing sodium azide, passed through sucrose gradients, and cryosectioned at 30 μm thickness. Blood samples were centrifuged at 14,000× *g* at 4 °C for 15 min and supernatant plasma was stored at −80 °C for future analysis.

### 2.3. ELISA

Brain: To prepare the mouse brain for anti-pGlu-3 Abeta 07/2a mAb ELISA analysis, the left hemisphere was homogenized by using a Precellys homogenizer (VWR) in TBS buffer (Thermo Fisher Scientific, Waltham, MA, USA) at a concentration of 150 mg brain per ml buffer containing protease inhibitor cocktail (Complete mini, Roche, Basel, Switzerland) and 0.1 mM AEBSF. The homogenate was centrifuged for 30 min at 25,000× *g* and supernatants were collected yielding the soluble TBS fractions. The protein concentration in the TBS fraction was determined using BCA Protein Assay Kit (Thermo Fisher Scientific, USA). To quantify 07/2a antibody concentrations in brain homogenates, streptavidin-coated 96-well plates were blocked with Pierce™ Protein-Free blocking buffer (Thermo Fisher Scientific, USA). for 2 h at RT and further treated with 20 ng of biotinylated pGlu3-Aβ3-17 peptide in each well at 4 °C for 2 h. After washing, antibody standards and brain samples were added to the plate and incubated at 4 °C for 2 h. The plates were washed three times and then incubated with anti-mouse IgG-HRP 500 ng/mL at 4 °C for 1 h. For detection, 100 µL SureBlue™ TMB substrate solution (KPL, Seracare, Milford, MA, USA) was added to each well for 30 min at RT and then the reaction was stopped by adding 50 μL of 1.2 N H_2_SO_4_. A SUNRISE Microplate Reader (Tecan, Männedorf, Switzerland) was used to measure optical density values at 450 nm. The resulting data were normalized to the concentration of extracted protein in the TBS fraction.

Plasma: The plasma from the mice was quantified for 07/2a mAb levels as follows. A streptavidin-coated plate was used to immobilize equivalent amounts of pE3-Aβ(3-17)-PEG-biotin. The diluted (1:50,000) plasma samples were then incubated directly in the wells thus prepared. After several washing steps, an incubation step with an anti-mouse Ab HRP-conjugate was performed. For detection, 100 µL SureBlue™ TMB substrate solution (KPL, USA) was added to each well for 30 min at RT and then the reaction was stopped by adding 50 μL of 1.2 N H_2_SO_4_. A SUNRISE Microplate Reader (Tecan, Switzerland) was used to measure optical density values at 450 nm and the data were normalized to the concentration of extracted protein in the TBS fraction.

### 2.4. Immunohistochemistry

DAB staining: For the mice that underwent unilateral FUS, brain sections were cut coronally and for the bilaterally FUS cohort sections were cut sagittally at 30 μm thickness. Four to 5 tissue sections at equidistant planes (700 μm apart) were collected. Anti-IgG2a immunostaining was performed to detect and quantify the 07/2a mAb immunoreactivity in the cortical and hippocampus regions. Briefly, tissue sections were quenched with 3% hydrogen peroxide for 15 min, blocked in 10% normal goat serum in PBS, followed by 4 °C overnight incubation on a shaker with biotinylated goat anti-mouse IgG2a (1:250, Southern Biotech, Birmingham, AL, USA; cat# 1081-08) followed by washes in PBS. Vectastain ABC kit (Vector Laboratories, Newark, CA, USA: PK-6100) was used to label biotin with HRP as per the manufacturer’s instructions and visualized with a 3,3′-Diaminobenzidine (DAB) staining kit (Vector Laboratories, USA; SK-4100). Sections were mounted onto glass slides, air-dried, and dehydrated before coverslipping with Permount. Colored images of the cortex and hippocampus were taken with a 10× objective on a Zeiss Axioimager microscope and the percent area of staining was quantified using Fiji Image J (1.53q, Java 1.8, NIH, Bethesda, Rockville, MD, USA).

Immunofluorescence staining: Free-floating coronal and sagittal brain tissue sections were washed in 1× PBS for 3 times 5 min each. After washes in PBS, sections were blocked in 10% normal goat serum for 30 min followed by 4 °C overnight incubation with primary antibodies against Iba1 (rabbit polyclonal, 1:1000; Dako, Wako, Osaka, Japan; cat# 019-19741), CD68 (rat monoclonal, 1:500, Serotec, Bio-Rad, Hercules, CA, USA; cat#MCA1957), K17 pGlu-3 Aβ-specific antibody (mouse IgG2b mAb, 1.12 mg/mL, gift of Vivoryon Therapeutics, Germany), S97 pan-Aβ (rabbit polyclonal, 1:5000, gift Dr. Dominic Walsh, BWH, USA), and Ly6G (rat monoclonal, 1:200, Biolegend, San Diego, CA, USA; cat#127602). The next day, sections were washed 3 times in PBS and incubated with Alexa-Fluor conjugated secondary antibodies goat anti-rat AF488, goat anti-mouse AF568, goat anti-rabbit AF647 (Invitrogen, Waltham, MA, USA; 1:500) for 2 h at room temperature. Following washes in PBS, sections were incubated in DAPI for 5 min, rinsed in PBS, mounted onto the slides, and coverslipped using PVA-DABCO aqueous mounting media.

Fluorescent stainings were visualized using a Nikon Ti Spinning Disk confocal microscope. Images were taken using a 20× objective with 20 µm z-stacks and no more than a 1 µm interval in cortex from 4–5 tissue sections per mouse. For every section, an average of 3–4 regions of interest (ROI) was taken while maintaining the same threshold across all the animals. Images were stacked and projected to maximum intensity and image analysis was performed using Fiji ImageJ. For the Ly6G staining, single-plane images were taken on Zeiss Axiovert with 10× objective using FITC filter. A custom-made Fiji ImageJ macro was used to measure the % area of immunoreactivity.

### 2.5. Microhemorrhage Detection

For Prussian blue staining, 4–5 sagittal brain sections 700 μm apart were stained using freshly prepared 2% potassium hexacyanoferrate trihydrate (Sigma, St. Louis, MO, USA; P3289) in 2% hydrochloric acid. The sections were incubated for 30 min followed by three washes in distilled water. For counterstain, sections were treated with Nuclear fast red (Vector Laboratories, USA; H-3403-500) for 5 min followed by alcohol dehydration in 95 and 100% ethanol for 5 min each. Tissue sections were cleared for 10 min in Histoclear and coverslipped with Permount.

### 2.6. Statistical Analyses

Statistics were performed using Graphpad Prism 9.0 (San Diego, USA). Normality of the residuals was analyzed using Anderson-Darling (A2*) test and differences in standard deviation through the Brown Forsythe test. Data were analyzed using analysis of variance (ANOVA) followed by Bonferroni’s multiple comparisons test. Data were expressed as mean ± SEM. For comparisons between hemispheres in a unilateral FUS study, a paired-*t*-test was used. Differences among groups were considered significant at values of *p* < 0.05.

## 3. Results

### 3.1. FUS Enhanced Targeted Delivery of 07/2a mAb

We performed an anti-pGlu3Aβ mAb infusion in combination with FUS and MB via sonication [33]. Twenty-four mo-old APP/PS1dE9 AD transgenic mice were sonicated either (i) unilaterally or (ii) bilaterally (Figure 1A) as described in the methods. To demonstrate BBB disruption, 1 h before euthanasia mice in the unilateral FUS group received an intravenous injection of Trypan blue, a dye that does not cross the barrier under normal conditions due to its size and hydrophilicity [34]. After unilateral sonication on the right hemisphere, we observed extravasation of the dye into the tissue, resulting in a visible blue area confirming BBB disruption (Figure 1A). The contralateral non-sonicated left hemisphere was identified with a small needle mark. To estimate the acute effects of focused ultrasound and whether 07/2a mAb was able to cross the BBB, we quantified cerebral levels of the antibody by ELISA after 4 h and 72 h post-treatment from the bilateral FUS treated mice. Our data shown in Figure 1B indicates a significant effect of 07/2a + FUS combination treatment (two-way ANOVA, F1,16 = 19.77, *p* < 0.05, mAb + FUS vs. mAb) with Bonferroni’s multiple comparison test indicating that the levels of 07/2a mAb in the hemibrain homogenates were significantly higher as early as 4 h (*p* < 0.005, mAb + FUS vs. mAb) and the levels remain significantly higher at 72 h (*p* < 0.05, mAb + FUS vs. mAb) post-treatment. No significant differences in the levels of antibodies were found within each treatment group between 4 h and 72 h time intervals (F1,16 = 1.048, *p* = 0.3).

The brain-to-blood ratio of the antibody reflecting the equilibrium across BBB demonstrated significantly increased levels of the antibody in the brain compared to the plasma compartment after the combination treatment (two-way ANOVA, F1,16 = 18.9, *p* < 0.005, mAb + FUS vs. mAb). Bonferroni’s multiple comparisons test showed a higher trend for brain-to-blood ratio of the antibody at 4 h (*p* = 0.1) and an increase by 72 h (*p* < 0.005) in mAb + FUS treated mice compared to mAb alone treated mice. Combination treatment resulted in an approximately 5-fold higher brain-to-blood ratio compared with antibody alone-treated mice indicating increased availability of the antibody in the brain after ultrasound treatment (Figure 1C). Although the brain-to-blood ratio of the antibody in the combined treatment group was increased at 72 h (*p* < 0.05), the difference from the increase at 4 h was not significant.

### 3.2. FUS Increased Antibody Immunoreactivity in Brain

We investigated the delivery of 07/2a mAb into the brain 4 h and 72 h after focused ultrasound treatment. Delivery of mAb to the sonicated right hemisphere in the unilateral FUS group was evaluated by anti-mouse IgG2a DAB immunohistochemistry (Figure 2A). Increased anti-IgG2a immunoreactivity was detected on the sonicated right hemisphere compared to the untreated contralateral left hemisphere and was restricted to the cortex, striatum, and to a lesser degree, the hippocampus. Quantification of the data revealed a significant increase in the cortical IgG2a antibody levels within the sonicated region at 4 h (df 4, t = 3.06, *p* < 0.05) and 72 h post-treatment (df 3, t = 3.22, *p* < 0.05, mAb + FUS vs. mAb; two-tailed, paired *t*-test, *n* = 4–5 mice per group) (Figure 2C). However, such changes in the hippocampal IgG2a immunoreactivity were not observed at either of the time points (Figure 2D). Triple immunofluorescence using Iba1, anti-IgG2a (for 07/2a mAb), K-17 (pGlu-3 Aβ IgG2b mAb) demonstrated that 07/2a mAb binds to the plaques surrounded by Iba1+ microglial cells (Figure 2B). In our bilateral FUS cohort, immunoreactivity in the cortex in 4 h and 72 h mice groups (Figure 2E,F) revealed a significant increase in IgG2a levels at 4 h in the mice that received combination treatment (*p* < 0.05, mAb + FUS vs. mAb alone, two-way ANOVA with Bonferroni post-hoc test) and an increasing trend at 72 h post-treatment (mAb + FUS vs. mAb alone) (Figure 2G). Anti-IgG2a immunostaining for 07/2a mAb in the hippocampus demonstrated an increasing trend of IgG2a at 4 h and 72 h, however, it was non-significant (Figure 2H). The increased presence of IgG2a immunostaining following intravenous infusion of the anti-pGlu3 Abeta mAb supports the increased delivery of 07/2a mAb into the brain.

### 3.3. FUS Increased the Microglial Immunoreactivity and Neutrophil/Monocyte Infiltration after Anti-pGlu3Aβ Treatment

To assess the acute inflammatory response in the brain after FUS exposure, coronal sections from the unilateral sonicated specimens and sagittal brain sections from the bilateral sonicated cohort were probed for the general microglial marker Iba1 and the phagocytic marker CD68. Brain images from the 4 h unilateral sonication group are shown in Figure 3A. Immunofluorescence analysis was performed for these markers in the cortical region in the mice that received 07/2a mAb alone, bilateral 07/2a mAb + FUS combination treatment, and 07/2a mAb combined with unilateral sonication, each group followed at 4 h and 72 h. In unilaterally sonicated mice, we observed a significant increase in Iba1 immunoreactivity in the sonicated right hemisphere at 4 h post-treatment compared to the non-sonicated contralateral left hemisphere (df 3, t = 3.75, *p* < 0.05). However, only an increasing trend in microglial Iba1 immunoreactivity was seen at 72 h (df 3, t = 3.07, *p* = 0.05). We also quantified the phagocytic activity of microglia in these cortical regions using the CD68 marker and found a significant increase in the CD68 staining on the ultrasound exposed side both at 4 h (df 3, t = 7.51, *p* < 0.005) and 72 h (df 3, t = 3.42, *p* < 0.05) post-treatment. In the bilateral FUS cohort, cortical Iba1 (F1,12 = 0.54, *p* = 0.87, two-way ANOVA) and CD68 (F1,12 = 1.97, *p* = 0.19, two-way ANOVA) immunoreactivity showed a non-significant increase in the mice that received combination treatment 07/2a mAb + FUS (bilateral) compared to the antibody alone treated mice 4 h and 72 h post-treatment. There was no significant effect of time (4 h vs. 72 h) on Iba1 immunoreactivity (F1,12 = 1.42, *p* = 0.26, two-way ANOVA) or CD68 immunoreactivity (F1,12 = 0.77, *p* = 0.39, two-way ANOVA) (Figure 3B).

To evaluate neutrophil/monocyte recruitment into the brain following antibody treatment and ultrasound exposure, coronal brain sections from the unilateral cohort were immunostained with the Ly6G marker (Figure 3C). Increased neutrophil staining was detected in the cerebral region 4 h post-treatment on the sonicated right hemisphere compared to the non-sonicated hemisphere (df 3, t = 3.34, *p* < 0.05). However, no such differences were observed after 72 h of treatment (df 3, t = 1.32, *p* = 0.28) (Figure 3D).

### 3.4. 07/2a + FUS Combination Treatment Does Not Increase the Number of Cerebral Microhemorrhages

Perls Prussian blue-stained brain sections from the bilateral and unilateral cohort were examined for the presence of microhemorrhages (Figure 4A). The number of cerebral microhemorrhages per brain section was quantified. After 4 h and 72 h post 07/2a mAb administration and unilateral sonication, no significant differences were observed between the non-sonicated left hemisphere vs. the sonicated right hemisphere at 4 h (df = 2, t = 0.66, *p* = 0.57) and 72 h (df = 3, t = 1.73, *p* = 0.18) post-treatment (Figure 4B). In the bilateral FUS cohort, there were no significant differences in the number of microhemorrhages deposits between the mice that received 07/2a mAb alone vs. 07/2a + FUS (bilateral) combination treatment (F1,12 = 0.0018, *p* = 0.90, two-way ANOVA) (Figure 4C) at either time points.

## 4. Discussion

The amyloid cascade hypothesis has been the rationale for the development of both active and passive immunization strategies to prevent or slow the progression of AD [35,36]. However, the delivery of antibodies or other drugs to the brain is limited by the presence of BBB, representing a major hurdle in treating neurological disorders [37]. The delivery of even small antibodies into the brain through passive diffusion is limited by the BBB and methods are under evaluation to circumvent or open up the barrier [38]. Recently, therapeutic ultrasound exposure has gained attention as a treatment modality for pathological aging including AD [39]. Advances in ultrasound technology and its use to enhance the delivery of therapeutic molecules across the BBB may be a valuable strategy in AD. For years pGlu-3 Aβ, due to its highly pathologic neurotoxic species that predominates N-terminally modified amyloid species, has been in limelight as a potential target for amyloid immunotherapy [40,41,42,43].

Previously, we showed that targeting pGlu-3 Aβ using 07/2a (anti-pGlu-3 Aβ mAb) in combination with FUS in a 3 weekly treatment paradigm led to a reduction in pGlu-3-42 Aβ and general Aβ plaque burden, increased glial activation and infiltrating neutrophils, and enhanced cognitive performance in an AD-like mouse model [33]. Importantly, we showed a significant increase in Ly6G cells associated with Ab plaques in the mice treated with both the mAb and FUS compared to the PBS alone and mAb alone groups, suggesting that FUS-BBBD in combination with the mAb faciliated neutrophil/monocyte infiltration into the brain. There were no significant differencess in the number of microhemorrhages between the four treatment groups, indicating that FUS did not increase the risk of microbleeds. In this present study, we focused on the acute effects (4 h and 72 h post-treatment) of FUS + MB and 07/2a mAb combination treatment. BBB opening after sonication is transient and supposedly closes within 24 h in wild-type rodents. Following ultrasound, a transient upregulation of inflammatory responses and its dampening by 24 h post-sonication has been reported [44]. Considering the very old age of these transgenic mice (24 months) which develop increasing vascular amyloid pathology with ageing, the closure of BBB after FUS treatment may be compromised and therefore less efficient at closing. Therefore, we have chosen 4 h as acute time window and 72 h time window for later response. Using FUS, we compared 07/2a mAb abundance in the brain and associated immune response in both sonicated and non-sonicated areas.

We found that 07/2a mAb + bilateral FUS combination treatment increased antibody levels approximately 5.6-fold in the brain at 4 h and 72 h following treatment compared to mice that received antibody alone. Increased antibody levels were detected at 4 h in the brain after combination treatment and maintained at 72 h, indicating the sustained presence of antibodies following sonication. Our immunohistochemistry data indicated an increasing trend for antibody staining at both time points. Particularly a significant increase in cortical 07/2a IgG2a immunoreactivity was evident following 4 h treatment. Mice that underwent 07/2a mAb administration & unilateral sonication on the right hemisphere showed a significant accumulation of the antibody on the sonicated hemisphere compared to the contralateral left hemisphere. We observed a significant increase in 07/2a IgG2a immunoreactivity in the cerebrum and minimal staining around the hippocampus. Quantification of 07/2a IgG2a immunoreactivity in the hippocampus on the sonicated hemisphere showed no changes by 4 h, however, an increasing trend was observed at 72 h, although non-significant. In our bilaterally FUS cohort, an increasing trend for 07/2a immunoreactivity was observed in the hippocampus at 4 h and 72 h.

To estimate the penetration capacity of the antibody from peripheral to central compartment following focused ultrasound exposure, the ratio of brain-to-blood antibody concentration was estimated from the half brain homogenates and their respective plasma from the bilaterally FUS cohort mice. We found a significant increase in the ratio of antibody levels in the brain compared to plasma, indicating an increased uptake into the brain following sonication reflecting the equilibrium across BBB [13] and probably increased presence of antibody around plaques.

In line with the previous literature showing an acute increase in proinflammatory cytokines and activated glia following sonication [45,46], we probed for the general microglia/macrophage marker Iba1 and the phagocytic marker CD68 in the brain and observed an overall increase in their immunoreactivity from 4 to 72 h in the mice that received combination treatment (mAb + bilateral FUS) compared to the mice that received 07/2a antibody alone. Such a slight increase in glial or phagocytic activity in this cohort could be probably explained by a low degree of BBB leakiness (cannot confirm) after sonication as estimated with lower levels of anti-IgG2a immunostaining seen in these sections compared to the IgG2a immunoreactivity seen in the unilateral sonicated cohort. In our unilaterally FUS cohort, the increase in Iba1 & CD68 immunoreactivity on the sonicated right hemisphere compared to the contralateral non-sonicated left hemisphere is strikingly apparent as early as 4 h and such response persisted even after 72 h post-treatment. Although, FUS exposure contributes to an increase in the penetration of endogenous IgG and IgM antibodies from the periphery to the brain and stimulation of immune responses and microglial activation [26,47], the significant increase in glial immunoreactivity could be mainly due to the increased 07/2a mAb abundance in the brain aided by FUS and the antibody-mediated microglial response to plaques. Indeed, in our recent study, we found a minimal increase in glial activation in ultrasound-alone treated mice [33]. FUS exposure also elevated the levels of infiltrating neutrophils into the brain as estimated by Ly6G staining, a marker specifically for neutrophils [48]. The increase in Ly6G immunoreactivity was seen only after 4 h post-treatment and not after 72 h. Such an effect can be explained in line with other ultrasound studies showing acute inflammatory response in the brain microvasculature transcriptome with increased expression of markers like Ccl2, Ccl3 by 6 h and C3, Ccl6, Gfap, Itgb2 by 24 h indicative of immune cell infiltration or migration [44]. We saw an increase in IgG2a immunoreactivity after 4 h and 72 h and an increase in neutrophil infiltration only at 4 h. It is possible that, as a first line of defense, there is an initial, transient rise in the peripheral infiltrating immune cells, as seen in acute inflammation, which may later be taken over by responses from the resident immune cells, such as microglia. Importantly, the absence of the expected increase in neutrophil recruitment/Ly6G+ staining at 72 h could be also due to their estimated half-life (average 12.5 h) and their short survival of 0.75 days in mice [49]. Also, repetitive FUS-induced BBB disruption and combination treatment for 3 weeks increases the recruitment of Ly6G positive immune cells and their association with the plaques [33]. Although neuroinflammation plays a central role in the pathogenesis of AD, transient inflammatory response in the endothelial membrane following FUS may aid plaque clearance. Infiltration of peripheral immune cells and activation of immune cells could contribute to lowering plaque load [44,50]. While studies showed recruitment of leukocytes begins during the FUS induced BBB-D [51] and others provide evidence for measuring at 5 min [45] and 6 h [44], our study provides and supports the evidence of transient neutrophil infiltration after 4 h and not after 72 h following FUS treatment, which may be independent of the 07/2a mAb.

Cerebral microbleeds are associated with cognitive decline and are an unwanted side effect in various anti-Aβ immunotherapies including the recently approved aducanumab. The risk of abnormalities ARIA-microhemorrhages (Amyloid related imaging abnormalities-Hemorrhages) and ARIA-edema identified on magnetic resonance imaging were detected in about 19% and 35% of the patients respectively in phase 2 and 3 clinical trials of aducanumab [7]. Prussian blue staining to detect hemosiderin (iron) was carried out to determine whether sonication and in combination with the treatment antibody caused microbleeds. While some Prussian blue deposits were observed in these aged 24 mo-old APP/PS1dE9 mice that received sonication on the right hemisphere (unilateral FUS) or combination treatment in the bilateral FUS cohort, we did not detect any significant increases with respect to their controls. Although some studies showed that FUS treatment has been associated with concurrent damage to the vasculature leading to microhemorrhages [52] and erythrocyte extravasation [53], owing to a short single-dose treatment and the acute effects thereupon, we did not see any significant increase in microbleeds after ultrasound exposure.

There are some limitations to this study. Due to the high mortality of these mice with aging, the number of mice used in each treatment group was limited. We only employed a few female mice in our unilateral cohort and none in the bilateral FUS cohort. While studies suggested sex-specific differences in endothelial function and BBB integrity, the inclusion of a larger cohort of both sexes will further delineate any sex-specific differences in immune cell trafficking following ultrasound exposure [54].

## 5. Conclusions

Taken together, our data support the idea that focused ultrasound can be considered an adjuvant therapy to enhance antibody delivery to the brain in treating AD. Indeed, our results suggest a 5.6-fold increase in 07/2a anti-pyroglutamate3Aβ mAb levels in the brain with activated microglia and transient infiltration of neutrophils by 4 h after ultrasound exposure. Therefore, increasing antibody delivery to the brain by FUS represents a possible strategy for transient BBB disruption without any signs of additional hemorrhages and may be considered a valuable therapeutic tool to support amyloid immunotherapy.

## Figures and Tables

**Figure 1 biomolecules-12-00951-f001:**
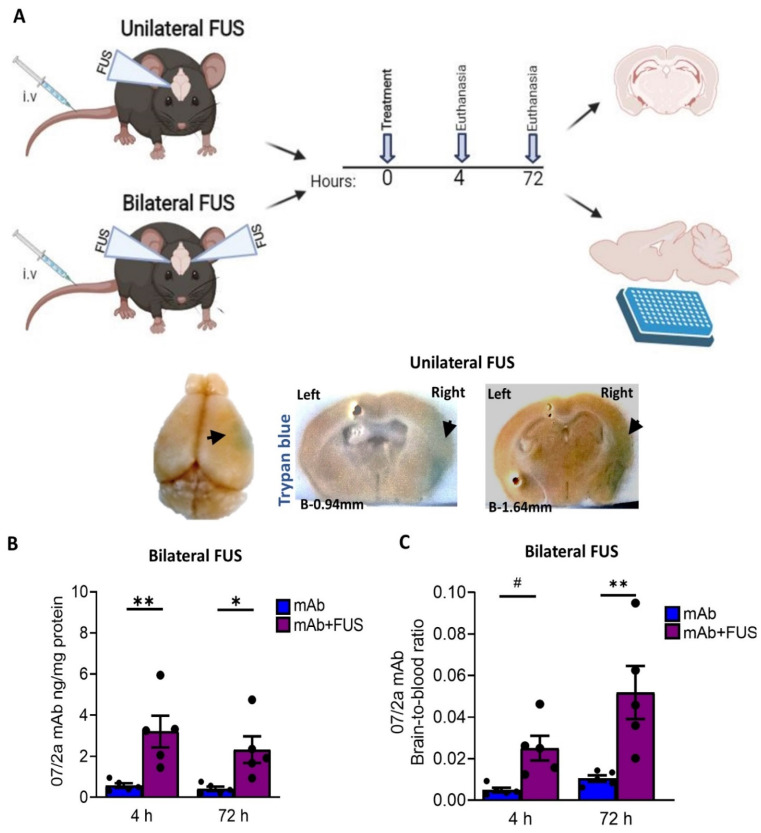
Study overview and the bioavailability of the 07/2a mAb in the brain (**A**) Experimental design and timeline of the experiment (image generated using BioRender). For unilateral FUS, 24 months old APP PS1dE9 mice were i.v. administered with single dose of 300 µg 07/2a mAb with microbubble i.v. infusion and FUS-BBBD on the brain’s right hemisphere. Mice received 2% Trypan blue dye prior to euthanasia and were taken down at 4 h (*n* = 5) and 72 h (*n* = 4) after mAb infusion; saline perfused and brains were collected for coronal sections. Non-sonicated left hemisphere was identified with a small needle mark. For Bilateral FUS cohort, both hemispheres received ultrasound and the same procedure has been followed but without Trypan dye injection. Collected brains were bisected; right hemispheres were used for ELISA and left hemisphere for histology. Representative pictures for trypan blue extravasation (arrows) on the sonicated right hemisphere across Bregma −0.94 mm to −1.64 mm from the unilateral FUS study. (**B**) Anti pGlu3-42 Aβ mAb levels from the half brain homogenates after 4 h and 72 h post treatment in the bilateral FUS cohort were analyzed by ELISA (**C**) Brain-to-blood concentration ratio of 07/2a mAb at 4 h and 72 h. Black dots represent data from individual mice. Data are expressed as mean ± SEM. Two-Way ANOVA, with Bonferroni’s multiple comparison test # *p* = 0.1, *, *p* < 0.05, **, *p* < 0.005.

**Figure 2 biomolecules-12-00951-f002:**
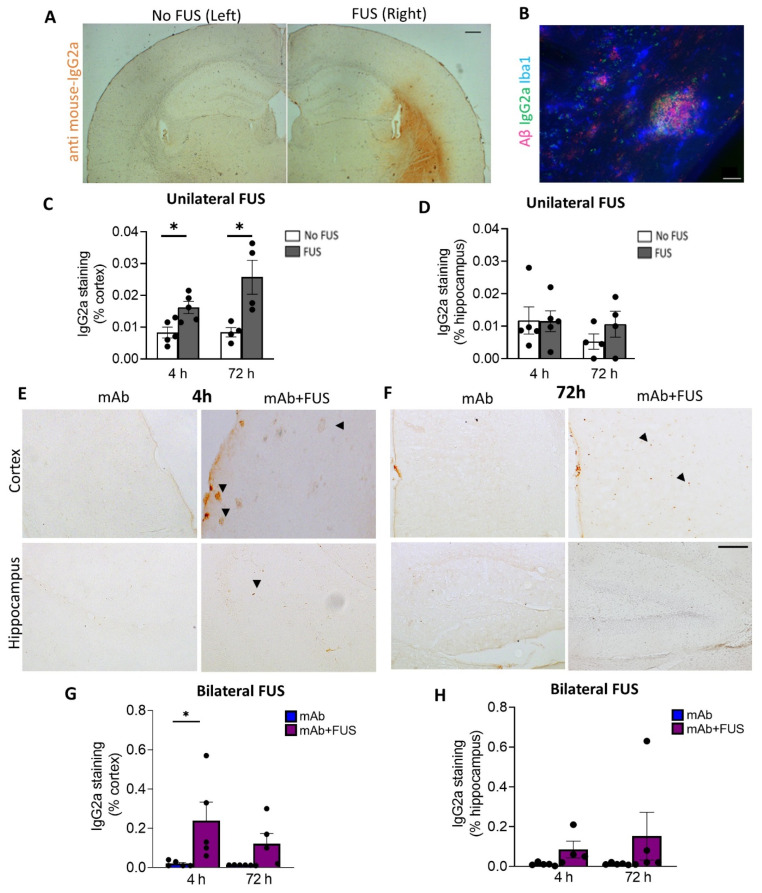
Focused ultrasound and IgG2a immunoreactivity (IR). (**A**) Representative DAB staining (anti mouse-IgG2a) for the left and right hemispheres in the unilateral FUS cohort. (**B**) Triple immunofluorescence labelling for Iba1, S97 (general Aβ) and anti-IgG2a indicating presence of antibody with Aβ plaques on the FUS side (right hemisphere). Unilateral FUS on the right hemisphere showed (**C**) increased anti-IgG2a IR in the cortex at 4 h (*p* < 0.05) and 72 h (*p* < 0.05) after 07/2a treatment and (**D**) no significant changes in the hippocampus; paired-*t*-test. (**E**,**F**) Cortical and hippocampal anti-IgG2a IR in the bilateral FUS cohort at 4 h and 72 h post-treatment. Arrowheads indicating IgG2a IR with some possible staining around plaques. Increased anti-IgG2a IR was found in the (**G**) cortex and (**H**) hippocampi after combination treatment at both time points, however, a significant increase was found in the cortex at 4 h. Black dots represent data from individual mice. Two-Way ANOVA, with Bonferroni’s multiple comparison test *, *p* < 0.05. Scale bar, (**A**) = 200µm and (**B**,**E**) = 20 µm.

**Figure 3 biomolecules-12-00951-f003:**
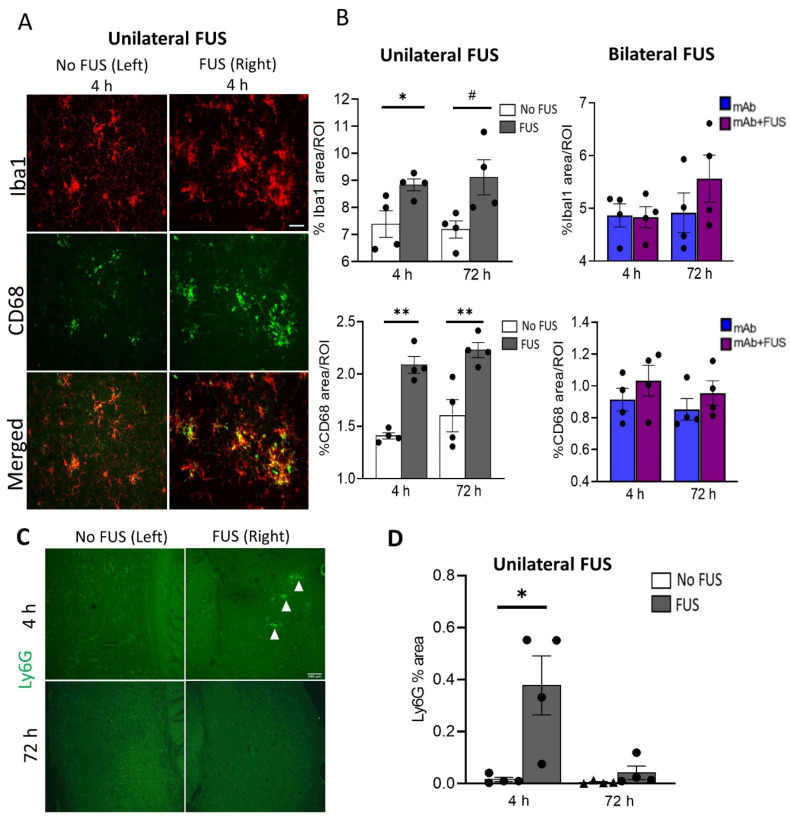
Acute immune response after single dose of 07/2a and FUS treatment. (**A**) Representative images (20×) for Iba1 and CD68 immunoreactivity after 4 h of 07/2a administration and unilateral sonication on the right hemisphere indicating increased microglial area coverage in the cortex compared to the non-FUS contralateral left hemisphere. (**B**) Quantitative analysis for general microglial marker (Iba1) and phagocytic marker (CD68) coverage in the cortical region from unilateral FUS cohort. After 07/2a mAb administration and sonication on the right hemisphere an increase in glial activity was seen compared to the contralateral left hemisphere (No FUS) control. Paired t-tests (two-tailed, α = 0.05). In the bilateral FUS cohort, 07/2a mAb treated alone or 07/2a + FUS treated mice after 4 h and 72 h post treatment showed a slight increase in glial activity. Data are represented as mean ± SEM. Two-Way ANOVA with Bonferroni’s multiple comparison test. (**C**) Representative pictures (4×) of Ly6G staining after 4 & 72 h treatment in the sonicated right hemisphere and contralateral non sonicated hemisphere. White arrowheads indicate Ly6G^+^ staining. (**D**) Area positive for Ly6G measured in the 4 h & 72 h unilateral FUS cohort expressed as percentage cerebral area, Paired *t*-tests (two-tailed, α = 0.05). Black dots represent data from individual mice. # *p* = 0.05, * *p* < 0.05, ** *p* < 0.005. Scale bar (**B**) = 50µm, (**D**) = 100 µm.

**Figure 4 biomolecules-12-00951-f004:**
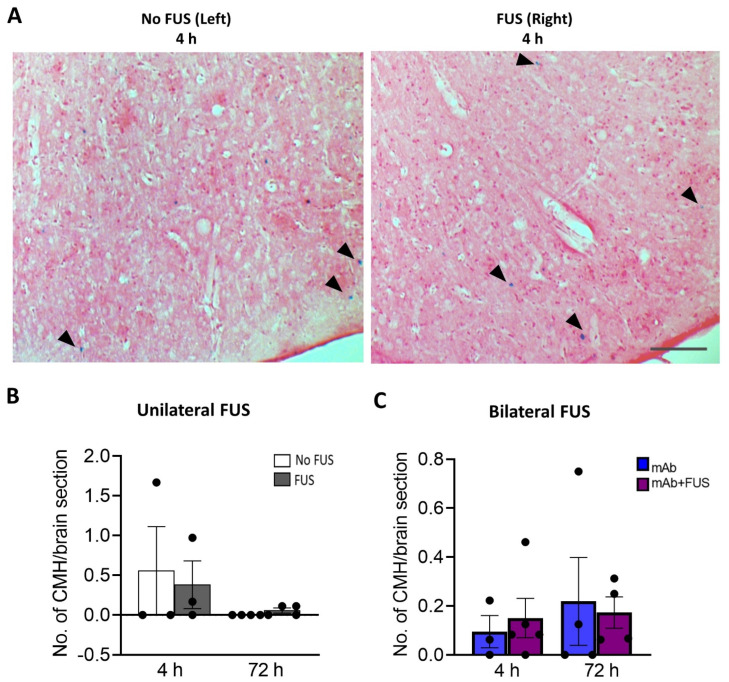
Prussian blue-positive cerebral microhemorrhages (CMH). (**A**) Representative pictures for the Perls’ Prussian blue staining in the cortex after unilateral sonication. Blue colored pigments (arrowheads) indicate hemosiderin deposits and these microhemorrhages are observed around capillaries. (**B)** Unilateral FUS cohort with sonication on right hemisphere did not show any significant microhemorrhages compared to the contralateral non sonicated left-hemisphere. Paired *t*-tests (two-tailed, α = 0.05). (**C**) In the bilateral FUS brain tissue sections no significant CMH were observed both at 4 h and 72 h post treatment. Two-way ANOVA with Bonferroni’s multiple comparison. Data are represented as mean ± SEM. *n* = 3–5 mice per group. Scale bar (**A**) = 200 µm.

## Data Availability

Data supporting the findings are available on request from the corresponding author.

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
