# Peer review of "Acute Effects of Focused Ultrasound-Induced Blood-Brain Barrier Opening on Anti-Pyroglu3 Abeta Antibody Delivery and Immune Responses"

_biomolecules, 2022, doi:10.3390/biom12070951_

Round 1

Reviewer 1 Report

The article submitted by Bathini et. al  investigated the effects of focused ultrasound to temporarily open the blood-brain barrier  and improve the delivery of antibodies in a transgenic mouse model.  The article was well written, appropriate controls were implemented, and the authors conclusions matched the presented data. 

I have several minor concerns:  1) Please ensure that all graph axis labels are consistent in font/size. 2) Please label Cortex/Hippocampus on the actual figure image(s) in Figure 2 C, D, G, H.  3)  Please describe what arrowheads are implicating in the Figure 2 legend for panels E, F. 4) In Figure 3 please consider presenting the images first and then the corresponding quantified graphs...the flow of the figure will be improved. 5) Figure 4, although critical to verify the absence of microhemorrhages seems out of place here.  Please consider  a restructure of the figures.  I suggest combining Figure 1A and Figure 4 as the first figure, and Figure 1B and C could be stand alone. I believe this would improve the flow of the manuscript without the microhemorrhages appearing to be an afterthought.

Thank you 

Reviewer 2 Report

Major comments

  1. The controls were missing: (1) The mice treated with only PBS without FUS (2) the mice treated with PBS+FUS. These two controls could solve the question, whether the inflammatory cells infiltration are solely based on the FUS transient opening the BBB or the antibody influx recruiting them. The other important point is to directly show the cerebral microhemorrhages in the mice treat with PBS only vs antibody+FUS combination. Only the comparison of antibody only vs antibody+FUS combination is not enough to support the safety of FUS combination, since several Alzheimer antibody clinical trials have already shown the ARIA (Amyloid related imaging abnormal), and which is a major concern for Alzheimer antibody targeting Abeta. 

  1. The experiment only showed the increase of microglia and neutrophil infiltration into the local CNS ROI, but didn’t prove the infiltration was either because of the FUS itself opening the BBB or the higher anti-Pyroglu3 Abeta antibody recruiting the inflammatory cells. The immunofluorescent colocalization of Abeta, anti-Pyroglu3 Abeta IgG2a, Iba1, and Ly6G should be shown for FUS only vs FUS+antibody at both 4h and 72h. This is an important point for the acute immune response.

  1. Since the Abeta was stained in the immunofluorescent picture, I’m surprised the researchers didn’t show the data to answer whether the antibody+FUS combination could start to clean the Abeta area at 4h or 72h.

Minor comments:

1.Figure 2A, scale bar is missing.

2. Please discuss the rationale to split the mice into two studies: unilateral FUS and Bilateral FUS. Considering unilateral FUS having both female and male mice, bilateral FUS only with male mice, bilateral FUS group should be more homogeneous. Why is the unilateral FUS group more likely to show a statistical difference, but bilateral FUS could not but only show a similar trend.

3. Please discuss the rationale to choose both 4h and 72h. Discuss the reason why some experiments show similar changes at 4h and 72h, but others show a totally different change (neutrophil increase at 4h, but disappeared at 72h). Is 4h defined as the acute phase, and 72h tends to be a consistent phase?

4. Figure 4 should show some representative prussian blue staining picture.

5. More needs to be explained why the immunohistochemistry focuses on cortical and hippocampus regions. Biomolecules journal readers are not all neurology specialists who might have limited Alzheimer’s disease pathology backgrounds.

Round 2

Reviewer 2 Report

Thank you for adding more introduction and discussion to the revised manuscript to clarify the paper for the reader. My questions have been fully answered. I think the manuscript has been improved to be published.